# Dispersion Measurement with Optical Computing Optical Coherence Tomography

**Wenxin Zhang** [1,2,3,†] **, Zhengyu Chen** [3,†]**, Xiao Zhang** [3,4]**, Chengming Wang** [3,5]**, Bin He** [3]**, Ning Liu** [1,2]**,**
**Yangkang Wu** [1,2]**, Yuxiu Tao** [1,2]**, Ning Zhang** [6] **and Ping Xue** [3,*]

1  College of Electronic and Optical Engineering, Nanjing University of Posts and Telecommunications, Nanjing 210083, China; zhangwx@njupt.edu.cn (W.Z.); liuning0127@njupt.edu.cn (N.L.); 1219023218@njupt.edu.cn (Y.W.); 1021020810@njupt.edu.cn (Y.T.)
2  College of Flexible Electronics (Future Technology), Nanjing University of Posts and Telecommunications, Nanjing 210083, China
3  State Key Laboratory of Low-Dimensional Quantum Physics, Department of Physics, Tsinghua University, Beijing 100084, China; chenzhen18@mails.tsinghua.edu.cn (Z.C.); zhangx@bit.edu.cn (X.Z.); wangchengming@jinsp-tech.com (C.W.); b-he19@mails.tsinghua.edu.cn (B.H.)
4  School of Life Science, Beijing Institute of Technology, Beijing 100081, China
5  Nuctech Company Limited, Beijing 100084, China
6  Institute of Forensic Science, Ministry of Public Security, Beijing 100038, China; zhangning@cifs.gov.cn
*  Correspondence: xuep@tsinghua.edu.cn
†  These authors contributed equally to this work.

**Abstract:** We propose a novel technique to measure fiber dispersion without any derivative operation and index measurement. Based on the relationship between the dispersion and the signal in optical computing optical coherence tomography, dispersion can be deduced with high accuracy from optical computing OCT signal position and resolution. The group velocity dispersion and third order dispersion of single mode fiber and dispersion compensating fiber with lengths of 10 m–10 km are measured to be in good consistence with the nominal value.

**Keywords:** dispersion measurement; optical computing; optical coherence tomography

## 1. Introduction

Dispersion parameter of fiber is quite important in the fiber industry. There are many methods to measure the dispersion. Most of them measure the refractive index and achieve the group velocity dispersion and third order dispersion with the derivative and second derivative of the index. The pulse delay method [1,2] measures the time of the light with different wavelengths to pass though the fiber to calculate the refractive index. As at each wavelength a monochromatic source is needed, the method is complex with the increase of the number of sampling point to achieve high accuracy. Phase shifting method [3] introduces an amplitude modulation after the light source and detects the modulation phase of the light after passing through the fiber. However, when the dispersion of fiber is very large, the difference of the phase and the reference phase will also be very large. In order to detect the exact phase, the step of the wavelength should be small enough, which may be costly and cumbersome. Interference method [4,5] can measure small dispersion, but when the length of the sample fiber is changed, the length of the reference has to, annoyingly, be changed. Using cross-correlation frequency resolved optical gating [6], a complex behavior of orthogonal polarization modes that is different in normal and anomalous dispersion regions of the photonic crystal fiber (PCF) can be revealed. However, this method is only suitable for PCF because it estimates PCF dispersion based on interplay between nonlinear and dispersive effects during supercontinuum generation in the PCF. Optical coherence tomography can also be used to measure the refractive index of a sample. Actually, it can even provide a precise index measurement of biological tissue [7]. However,

due to the short coherence length of OCT light source, the sample cannot be too large. This is also a problem with conventional OCT dispersion measurements. Method of fingerprint-spectral wavelength-to-time mapping can measure the dispersion with high accuracy, but it employs mode-locked fiber laser as the light source, which is costly [8], and has a fiber length limit of 0.5–100 km. Another method using transfer function [9] is developed to indirectly measure only the third order dispersion via a sequence of iterations and needs a double-sideband suppressed carrier, variable optical delay line, optical spectrum processor and network analyzer to measure the dispersion, which is quite cumbersome, high-cost and time-consuming. While in many cases, what is really concerned is the general dispersion properties of media such as long-distance optical fibers over a wide wavelength range.

Up until now, there has been a lack of technique enabling fast and simple measurement of both the group velocity dispersion and third order dispersion of large dynamic range with high precision by utilizing a single cheap light source.

In this letter, we propose a novel technique to obtain all the dispersion parameters with no need for an ultra-wideband laser or mode-locked fiber laser. Also, by using optical computing optical coherence tomography (OC-OCT or OC$^2$T) [10], we do not need any derivative operation and many index measurements. In this method, the fiber under test (FUT) is not placed in the interferometer and therefore its length has no theoretical limit. As a result, our method has large range of measurement. Therefore, we develop a new method based on OC$^2$T to measure dispersion for the first time. Our method provides a low-cost and simple dispersion measurement method with relatively high accuracy.

Unlike conventional OCT, OC$^2$T utilizes the dispersion as the method to implement optical computing to get the imaging signal and restrain the conjugate signal as well [11]. Therefore, the dispersion and the signal have an intrinsic relationship in OC$^2$T. This implies a new feasibility to measure the dispersion based on OCT signal. Later, we will firstly illustrate the relationship of the fiber dispersion and the signals in OC$^2$T and give a simple way to measure the dispersion.

## 2. Methods

The optical design of the OC$^2$T is shown in Figure 1. In the experiment, we use an LED as the light source with a center wavelength of 1550 nm and FWHM of 40 nm. The bandwidth of BD is 20 GHz. The complex amplitude of light from SLD is expressed as:

$$
\begin{aligned}
E_{SLD} &= S(\omega)\exp(-i\omega t) \\
&= \sqrt{S_0}\exp[-\gamma^2(\omega-\omega_0)^2/4]\exp(-i\omega t)
\end{aligned} \tag{1}
$$

where $t$ is the time variable, $\omega$ the frequency of light, $\omega_0$ is the center frequency and $S_0$ is a constant. $\gamma = (2\ln2)^{1/2}\lambda_0^2/(c\pi\Delta\lambda)$ describes the bandwidth of the light in angular frequency and $\Delta\lambda$ is the FWHM bandwidth in wavelength.

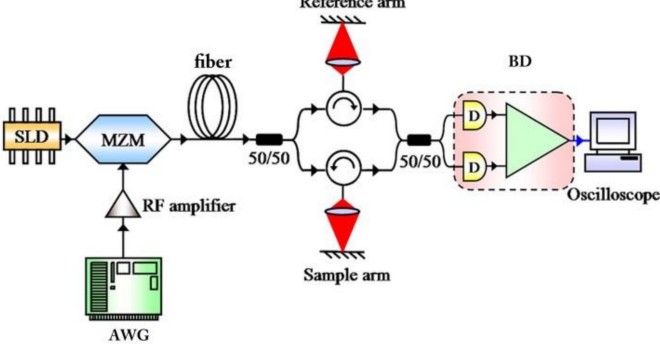

**Figure 1.** Optical design of the OC$^2$T. SLD: super luminescent diode, MZM: Mach–Zehnder modulator, AWG: arbitrary waveform generator, RF: radio frequency, BD: balanced detector.

An MZM is used to modulate the light. The modulation function $m(t)$ is generated by an arbitrary waveform generator (AWG). The signal after MZM is:

$$E_{MZM}(\omega, t) = m(t)E_{SLD}, \qquad (2)$$

where $m^2(t) = [1 + \cos(at^2)]/2$ and $a$ is a constant. In the experiment, the parameter $a$ is $1.6 \times 10^{-17} \, \text{s}^{-2}$.

A section of fiber is used as dispersion media after MZM. We define the optical pathlength:

$$\begin{aligned} L' &= Ln = L(n_0 + \beta_L \omega' + \beta_{L2} \omega'^2), \\ \omega' &= \omega - \omega_0. \end{aligned} \qquad (3)$$

where $L$, $n$ and $\omega_0$ are fiber length, refractive index and center frequency of light, respectively. While $n_0$, $\beta_L$ and $\beta L_2$ are refractive index at $\omega_0$, group-velocity dispersion and third order dispersion of the fiber, respectively.

The signal after fiber is:

$$\begin{aligned} \left|E_L(\omega,t)\right|_2 &= m^2(t - L'/c)\left|s(\omega)\right|^2 \\ &= \tfrac{1}{2}S(\omega)\left\{1 + \cos[(at'^2) + (-2at'\tfrac{L}{c}\beta_L)\omega' + (a\tfrac{L^2}{c^2}\beta_L{}^2 - 2at'\tfrac{L}{c}\beta_{L2})\omega'^2]\right\} \\ &= \tfrac{1}{2}S(\omega)[1 + \cos(\alpha_1 + \alpha_2 \omega + \alpha_3 \omega^2)] \\ &where \;\; t' = t - n_0 L/c, \; \alpha_1 = at'^2, \alpha_2 = -2at'\tfrac{L}{c}\beta_L, \\ &\alpha_3 = a\tfrac{L^2}{c^2}\beta_L{}^2 - 2at'\tfrac{L}{c}\beta_{L2}; \;\; t' \in [-\tfrac{T}{2}, \tfrac{T}{2}] \end{aligned} \qquad (4)$$

$T$ is the duration of the modulation signal $m(t)$.

Therefore, the signal passes through the interference and reaches the balanced detector [9]:

$$\begin{aligned} I(t') &= \int \int 1/2 \times S(\omega) r_s(z) r_r \sin(\beta_2 \omega' + \beta_3 \omega'^2) dz d\omega' \\ &+ \int \int 1/2 \times S(\omega) \cos(\alpha_1 + \alpha_2 \omega' + \alpha_3 \omega'^2) \\ &\quad \times r_s(z) r_r \sin(\beta_2 \omega' + \beta_3 \omega'^2) dz d\omega' \\ \beta_2 &= 2(\Delta l/c + n_s z/c) \\ \beta_3 &= 2\beta_l l/c \end{aligned} \qquad (5)$$

$n_s$ represents the sample refractive index, $\Delta l$ the optical path difference, $l$ the length of the dispersion media and $\beta_l$ the group velocity dispersion of the insert media. $r_s$, $r_r$ represent the reflectivity of the sample and reference arm, and $z$ the distance between the imaging point and the surface of the sample.

As a mirror is used as the sample, i.e., $z = 0$, we have:

$$\begin{aligned} \beta_2 &= 2\Delta l/c, \beta_3 = 2\beta_l l/c, \\ r_s(z) &= \delta(z = 0) \\ I(t') &= \int 1/2 \times S(\omega) r_r \sin(\beta_2 \omega' + \beta_3 \omega'^2) d\omega' \\ &+ \int 1/2 \times S(\omega) \cos(\alpha_1 + \alpha_2 \omega' + \alpha_3 \omega'^2) \times r_r \sin(\beta_2 \omega' + \beta_3 \omega'^2) d\omega' \end{aligned} \qquad (6)$$

We neglect the DC background in Equation (6) and rewrite the signal of the OC$^2$T system as:

$$\begin{aligned} I'(t') &= \tfrac{1}{2} r_r \int S(\omega) \cos(\alpha_1 + \alpha_2 \omega' + \alpha_3 \omega'^2) \times \sin(\beta_2 \omega' + \beta_3 \omega'^2) d\omega' \\ &\propto \frac{S_0 \cos\left\{\alpha_1 - [\frac{(\alpha_2 - \beta_2)^2(\alpha_3 - \beta_3)}{\gamma^4 + 4(\alpha_3 - \beta_3)^2}] + \theta_-\right\}}{\sqrt[4]{1 + 4(\alpha_3 - \beta_3)^2/\gamma^4}} \times \exp\left\{\frac{-(\alpha_2 - \beta_2)^2}{2\gamma^2[1 + 4(\alpha_3 - \beta_3)^2/\gamma^4]}\right\} \\ &+ r_s(z) \frac{S_0 \cos\left\{\alpha_1 - [\frac{(\alpha_2 + \beta_2)^2(\alpha_3 + \beta_3)}{\gamma^4 + 4(\alpha_3 + \beta_3)^2}] + \theta_+\right\}}{\sqrt[4]{1 + 4(\alpha_3 + \beta_3)^2/\gamma^4}} \times \exp\left\{\frac{-(\alpha_2 + \beta_2)^2}{2\gamma^2[1 + 4(\alpha_3 + \beta_3)^2/\gamma^4]}\right\} \\ &where \;\; \theta_\mp = \tan^{-1}[2(\alpha_3 \mp \beta_3)/\gamma^2]/2 \end{aligned} \qquad (7)$$

As shown in Equation (7), the signal appeared at $\alpha_2 = \pm\beta_2$, the first term appeared at $\alpha_2 = \beta_2$ may be defined as the signal, and thus the second term appeared at $\alpha_2 = -\beta_2$ as its conjugate. As $\alpha_2$ is in direct proportion to the group-velocity dispersion, the value of group dispersion will be reflected in the position of the signal. Hence:

$$\begin{aligned} \alpha_2 = \beta_2 &\rightarrow t' = -\frac{\Delta l}{aL\beta_L} \\ \alpha_2 = -\beta_2 &\rightarrow t_c' = \frac{\Delta l}{aL\beta_L} \end{aligned} \tag{8}$$

Based on Equation (8), the signal position $t'$ is proportional to $\Delta l$, and the scale factor is associated with the group-velocity dispersion of the fiber. Therefore, we can calculate the group-velocity dispersion by measure the relationship between $\Delta l$ and $t'$:

$$\beta_L = -\frac{1}{aL}\frac{d\Delta l}{dt'}. \tag{9}$$

Now we consider the axial resolution of the signals. Because there is no dispersion difference between the sample arm and reference arm, we have $\beta_3 = 0$ and the resolution is affected by $\alpha_3$. As the value of $\alpha_3$ is a function of the third order dispersion $\beta_{l2}$ and the position of the signal $t'$, the signal at different $t'$ will have different resolution, and the change of resolution is determined by $\beta_{l2}$. As both signals are Gaussian shape, the axial resolution of the signal and its conjugate are their FWHMs:

$$\delta z' = \delta z_c' = \delta z'_0 \sqrt{1 + 4\alpha_3^2/\gamma^4}, \tag{10}$$

where $\delta z_{0'} = (2\ln2)\,\lambda_0^2/(\pi\Delta\lambda)$ is exactly the axial resolution of traditional OCT

Based on Equation (10), the resolution of OC$^2$T changes with $\alpha_3$ and $\alpha_3$ is a variable of $t'$ and third order dispersion $\beta_{L2}$. To measure the resolution, we can calculate the third order dispersion of the fiber.

$$\begin{aligned} \left|\alpha_3\right| &= \frac{\gamma^2}{2}\sqrt{\left(\frac{\delta z'}{\delta z'_0}\right)^2 - 1} \\ \beta_{L2} &= -\frac{c}{2aL}\frac{d\alpha_3}{dt'} \end{aligned} \tag{11}$$

Although we can only calculate the absolute value of $\alpha_3$ from Equation (11), the sign is easy to determine, because $\alpha_3$ linearly changes with $t'$ and is always positive at $t' = 0$. Therefore, we can calculate the third order dispersion by Equation (11).

The flow chart is shown in Figure 2. The fiber under test is placed after MZM. We then change the optical path difference between the reference arm and sample arm $\Delta l$ and record the signal position $t'$ and resolution $\delta z'$. Finally, we can calculate the group-velocity dispersion and third order dispersion by Equations (8) and (11).

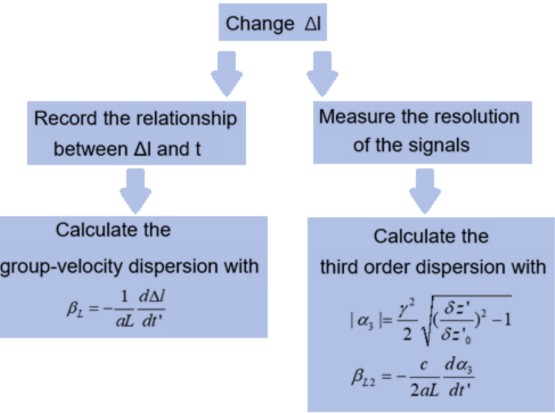

**Figure 2.** The process of measuring dispersion with OC$^2$T.

### 3. Experiments Result

We measure the dispersion parameter of 10 km of single mode fiber (SMF, $\beta_L = 6.5 \times 10^{-18}$ s, $\beta_{L2} = 2.2 \times 10^{-32}$ s$^2$) to prove the ability of our method.

In order to verify the experiment result, we first use the same parameters of our experiment to do the simulation, as shown in Figure 3. We measure the broadening of the simulation signals and find that $[(\delta z'/\delta z_{0'})^2 - 1]^{1/2}$ is linear with $t'$, which is consistent with the theory.

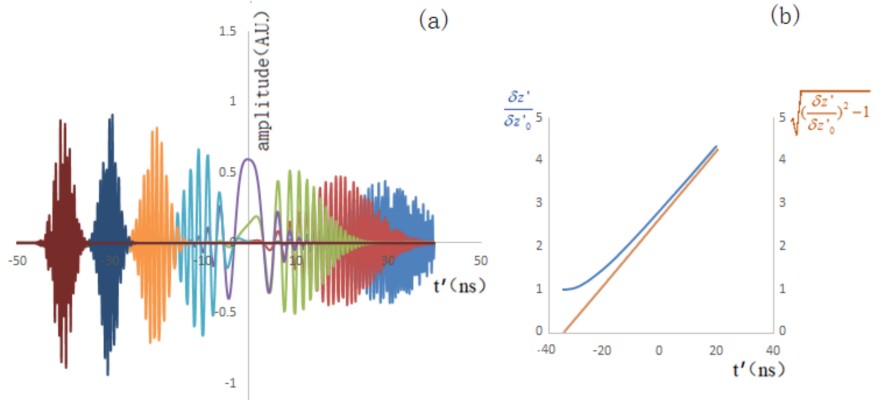

**Figure 3.** The simulation result using the parameter of 10 km SMF. (**a**) The signal at different $t'$, colors from left to right are used for signals at $t' = -40, -30, -20, -10, 0, 10, 20, 30$ ns, respectively; (**b**) the relationship between the resolution expansion and $t'$.

Figure 4 shows the experiment result, where we measured the signal at different $t'$ by changing the optical path difference $\Delta l$. In Figure 4a, we can see that the position and the width of the signal are changing with the optical path difference $\Delta l$. In Figure 4b,c, we prove experimentally that the optical path difference $\Delta l$ and $\alpha_3$ are linear with $t'$, which is in consistent with the theoretical results. We find the relationship between $\Delta l$ and $t'$ in Figure 4b and calculate the slope $d(\Delta l)/dt' = -(10.9 \pm 0.6)$ μm/ns. The group-velocity dispersion can be deduced from Equation (7):

$$\begin{aligned} \beta_L &= -\frac{1}{aL}\frac{d\Delta l}{dt'} \\ &= -\frac{1}{0.16\text{ns}^{-2} \times 10{,}000\text{m}} \times (-10.9 \pm 0.6 \text{μm/ns}) \\ &= (6.8 \pm 0.4) \times 10^{-18}\text{s} \end{aligned} \tag{12}$$

The third order dispersion can also be calculated by measuring the resolution broadening. We measure the resolution and calculate $2 \, |\, \alpha_3 \, |\, /\gamma^2 = [(\delta z'/\delta z_{0'})^2 - 1]^{1/2}$, as shown in Figure 4c. The slope $2/\gamma^2 * d(\alpha_3)/dt' = -(0.084 \pm 0.004)$ ns$^{-1}$. Therefore, the third order dispersion can be obtained from Equation (8):

$$\begin{aligned} \beta_{L2} &= -\frac{c}{2aL}\frac{d\alpha_3}{dt'} \\ &= -\frac{3*10^8\text{m/s}}{2 \times 0.16\text{ns}^{-2} \times 10{,}000\text{m}} \times \frac{1}{2 \times 1.776 \times 10^{26}\text{s}^{-2}} \times (0.084 \pm 0.004)\text{ns}^{-1} \\ &= -(2.2 \pm 0.1) \times 10^{-32}\text{s}^2 \end{aligned} \tag{13}$$

We also measure the dispersions of some dispersion compensating fibers (DCF) with different lengths (DCF-G.655/250, WuHan ChangFei, $\beta_L = -1.4 \times 10^{-16}$ s, $\beta_{L2} = -3.2 \times 10^{-30}$ s$^2$) to further evaluate our method experimentally, as shown in Figure 5. We can see that our method can measure the dispersion of short fibers even if the fiber is only 10 m long.

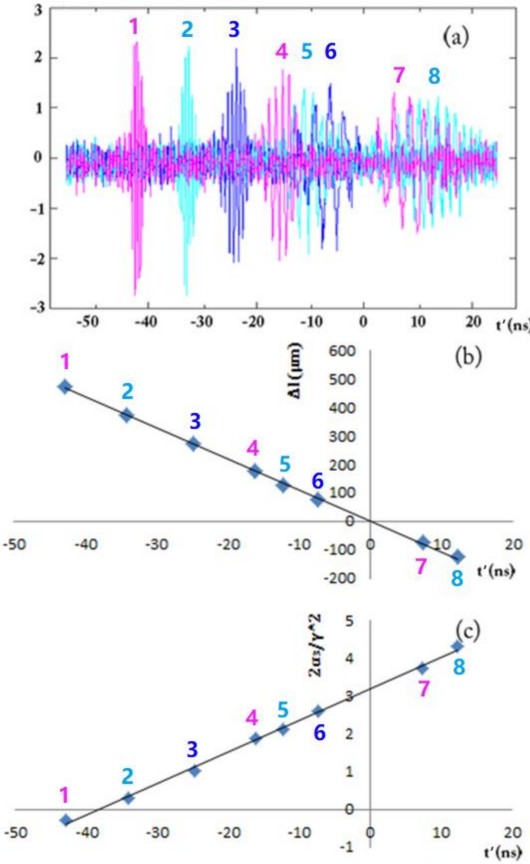

**Figure 4.** The measurement of the dispersion of 10 km SMF. (**a**) The signal at different $t'$, different colors and numbers are used for signals at different $t'$; (**b**) the relationship between the optical path difference $\Delta l$ and $t'$, the linearity between $\Delta l$ and $t'$ is 0.999; (**c**) the relationship between the resolution expansion and $t'$, the linearity between $\alpha_3$ and $t'$ is 0.999.

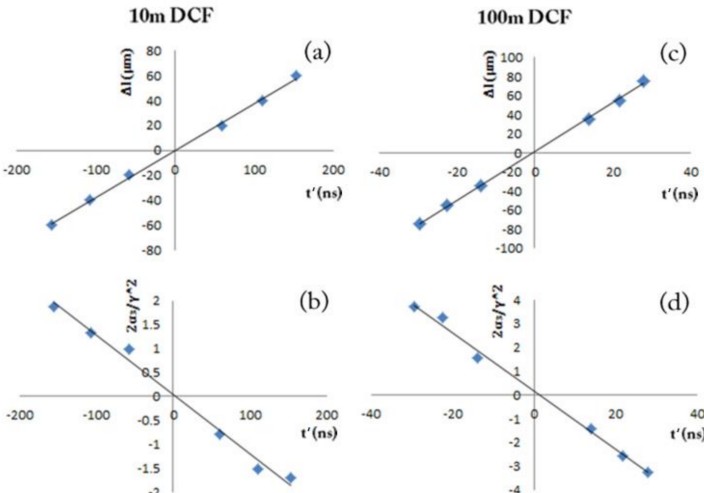

**Figure 5.** The measurement of the dispersion of 10 m and 100 m DCF. (**a**) The relationship between the optical path difference $\Delta l$ and $t'$ of 10 m DCF, the linearity between $\Delta l$ and $t'$ is 0.999; (**b**) the relationship between the resolution expansion and $t'$ of 10 m DCF, the linearity between $\alpha_3$ and $t'$ is 0.99. (**c**) The relationship between the optical path difference $\Delta l$ and $t'$ of 100 m DCF, the linearity between $\Delta l$ and $t'$ is 0.999; (**d**)the relationship between the resolution expansion and $t'$ of 100 m DCF, the linearity between $\alpha_3$ and $t'$ is 0.99.

As our method is computationally simple, we can calculate the slope by LINEST function in Excel. The LINEST function uses the least square method.

The comparison of dispersion parameters measured with our method and the theoretical value was shown in Table 1. In the table, the measurement results of our method are in good agreement with the factory specifications.

**Table 1.** Comparison of the experimental and factory specifications.

| | Measured $\beta_1$ (s) | Factory Specifications $\beta_1$ (s) |
|---|---|---|
| SMA 10,000 m | $(6.8 \pm 0.7) \times 10^{-18}$ | $6.5 \times 10^{-18}$ |
| DCF 100 m | $(1.6 \pm 0.9) \times 10^{-16}$ | $1.4 \times 10^{-16}$ |
| DCF 10 m | $(2.4 \pm 1.1) \times 10^{-16}$ | $1.4 \times 10^{-16}$ |
| | Measured $\beta_{l2}$ (s$_2$) | factory specifications $\beta_{l2}$ (s$_2$) |
| SMA 10,000 m | $(2.2 \pm 0.1) \times 10^{-32}$ | $2.2 \times 10^{-32}$ |
| DCF 100 m | $(3.2 \pm 0.1) \times 10^{-30}$ | $3.2 \times 10^{-30}$ |
| DCF 10 m | $(3.3 \pm 0.2) \times 10^{-30}$ | $3.2 \times 10^{-30}$ |

## 4. Discussion

As shown in Table 1, both the group velocity dispersion and the third order dispersion are well in agreement with nominal value. Also, we can find that relative uncertainty is smaller if the sample has larger dispersion $\beta_l * L$ or $\beta_{l2} * L$, i.e., longer length or larger dispersion parameter.

The instrument cost and complexity of our method are far less than those of pulse delay method, but our method can achieve comparable accuracy when we measure 10,000 m SMA. Although our method has relatively large error when measuring the group-velocity dispersion of short fibers, similar to other methods such as the pulse delay method, we can still measure the third order dispersion with higher accuracy than conventional methods even if the fiber is only 10 m long.

The measurement accuracy of the group-velocity dispersion mainly depends on the uncertainty of each point to measure $\Delta l$ and $t'$. Because $\Delta l$ is changed by a translation stage which has an accuracy of as high as 1 μm, its uncertainty can be neglected. The uncertainty of the signal position $t'$ is half of the resolution. Therefore, we have:

$$\delta \beta_L = -\frac{1}{aL} \left( \frac{\Delta l}{\Delta t' - \delta t'} - \frac{\Delta l}{\Delta t' + \delta t'} \right), \tag{14}$$

The uncertainty of the third order dispersion mainly comes from the measurement error of resolution broadening and $t'$. As the uncertainty of resolution broadening is affected by the distortion of modulation and signal detection, which is complicated to measure, we thus use the standard deviation of the slope of linear regression as an estimate. So, we have:

$$\Delta \beta_{L2} = -\frac{c}{2aL} \sigma_{\left( \frac{d\alpha_3}{dt} \right)}. \tag{15}$$

Compared with the traditional methods, our measurement system is simple and easy to set up. Also, both group velocity dispersion and the third order dispersion can be measured, and the results are well in agreement with nominal value. Our method is more compact and low-cost than conventional methods such as the pulse delay method and interference method because they employ a lot of mono-chromatic sources or reference arms with different lengths. Also, we obtain the group-velocity dispersion and the third order dispersion by calculating the slope of the signal position and resolution at different $t'$. Therefore, even if some measuring points have large error, the slope does not change that much. This implies that the measurement with our method is more robust and has good stability.

The comparisons between our method and some conventional methods are shown in Table 2.

**Table 2.** Comparison between our method and some conventional methods.

|  | Cost | Compactness | Accuracy (Group-Velocity Dispersion) | Accuracy (Third-Order Dispersion) | Measurement Range |
|---|---|---|---|---|---|
| our method | low | High | relatively high | high | large |
| pulse delay method | high | Low | relatively high | relatively high | relatively large |
| interference method | high | relatively high | high | high | small |

The phase shifting method has a limited dynamic range caused by the need of a single period of the phase modulation, while the interference method has its dynamic range limit around the length of the reference arm. Our method can measure the dispersion of fibers of any lengths, because the fiber under test (FUT) is not placed in the interferometer and therefore its length has no theoretical limit. However, the minimum value which is suitable for measurement is limited by the measurement accuracy. As already discussed, when measuring the group-velocity dispersion of short fibers, our method can still implement measurements, but with relatively large errors. Therefore, our method is more suitable for measuring samples with group-velocity dispersion larger than 10 m DCF. However, as for the third order dispersion, our method can obtain reliable results even if the fiber is only 10 m long.

In the following study, we will further improve the measurement accuracy and enable the method to measure even smaller dispersion. The SNR of the $OC^2T$ system will be further improved so that the position and resolution of the signal can be more precise. As a result, the standard deviation of the slope of linear regression will be smaller and the measurement accuracy can be further improved.

## 5. Conclusions

In conclusion, we develop a new method based on $OC^2T$ to measure dispersion. We illustrate signal of $OC^2T$ and analyze the relationship between the signal position, resolution and the dispersion in the $OC^2T$ system. Using these relationships, we demonstrate a novel method to measure the group velocity dispersion and third order dispersion of the fiber. Compared to the traditional methods, our measurement system is simple and easy to set up. Dispersion measurements of SMF and DCF with different lengths are also implemented. Both the group velocity dispersion and the third order dispersion are well in agreement with theoretical value.

**Author Contributions:** Conceptualization, W.Z. and P.X.; methodology, Z.C. and B.H.; validation, X.Z. and P.X.; formal analysis, C.W. and W.Z.; investigation, Z.C. and Y.W.; resources, P.X.; data curation, N.Z. and N.L.; writing—original draft preparation, W.Z.; writing—review and editing, Z.C. and P.X.; visualization, Y.T.; supervision, P.X.; project administration, P.X.; funding acquisition, P.X. All authors have read and agreed to the published version of the manuscript.

**Funding:** Supported by National Natural Science Foundation of China (61975091,61575108,61905015, 61227807); Beijing Natural Science Foundation (4194089); Open Research Fund Program of the State Key Laboratory of Low-Dimensional Quantum Physics (KF201908,KF202003); Nanjing University of Posts and Telecommunications, NY219148 (XK1060919148); The Foundation of Jiangsu Provincial Double-Innovation Doctor Program grant (CZ106SC20026); Beijing Institute of Technology Research Fund Program for Young Scholars (Grant No. 3160011181805); Open Project of National Engineering Laboratory for Forensic Science (2020NELKFKT01); Beijing Nova Program of Science and Technology (Z191100001119039).

**Informed Consent Statement:** Not applicable.

**Data Availability Statement:** Data underlying the results presented in this paper are not publicly available at this time but may be obtained from the authors upon reasonable request.

**Acknowledgments:** The authors thank colleagues in State Key Laboratory of Low-dimensional Quantum Physics, Department of Physics, Tsinghua University for valuable discussion.

**Conflicts of Interest:** The funders had no role in the design of the study; in the collection, analyses, or interpretation of data; in the writing of the manuscript, or in the decision to publish the results.

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
