# Peer review of "Dispersion Measurement with Optical Computing Optical Coherence Tomography"

_photonics, doi:10.3390/photonics9010048_

Round 1

Reviewer 1 Report

This paper describe the use of optical computing OCT to measure the fiber dispersion. The structure of the paper is clear. The idea is novel and the theoretical and experimental results are convincing. The paper could be considered to be published in the Journal Photonics. However, the following concerns should be addressed before further publication process.
(1) The main concern is that conventional fiber dispersion measurement methods are matured and widely used in the industry for years. More discussion on the distinct advantage of the proposed method should be given to show that the method is significant. For example, cost, compactness, robustness, etc. should be discussed in more details.
(2) The error of measurement with the proposed method should be compared with traditional method.
(3) The dynamic range of the proposed method should be evaluated and discussed.
(4) In the abstract, "Based on, ..." the sentence should be revised.
(5) In Eq. (1), \omega^2 should be (\omega - \omega_0)^2, where \omega_0 is the center frequency.
(6) In Fig. 2(a), there should be explanation on the different line colors.
(7) The English should be further polished.

Reviewer 2 Report

*       The significant trends of the simulation results should show.

*       Comparison with recent studies and methods would be appreciated.

*       Introduction section can add the issues in the current work context and how proposed algorithms/approaches can overcome this.

*       Literature review techniques have to be strengthened by including the current system's issues and how the author proposes to overcome the same.

*       Clarify the finding Error rate and accuracy in the performance analysis section.

*       It is suggested to add the chart for the given process with a description.

*       The mapping process for the proposed technique should be discussed in detail.

*       Conclusion should state scope for future work.

*       Authors should add more information on the code's implementation to perform the analysis and the library involved in this task.

*       Authors should add the parameters of the process/method.

*       The paper does not clearly explain its advantages concerning the literature: the novelty and contributions of the proposed work are not clear: does it offer a new method? Or does the innovation only consists of the application?

The advantage of the proposed method concerning other ways in the literature should be clarified.

*       The paper does not provide critical experimental details needed to assess its contribution: What is the validation procedure correctly?

*       The comparison of different methods using clear graphs should be explained.

*       Results need explanations. Additional analysis is required for each experiment to show its primary purpose.

Round 2

Reviewer 1 Report

I have reviewed the paper previously, the revised paper has addressed all of my concerns. So I suggest acceptance of the revised paper.

Reviewer 2 Report

The authors have done all the suggested corrections, Now the paper is ready for publication.